# A Feasibility Study of a Fit Kit School-Based Intervention to Improve the Health of Students and Their Families

**Jenna M. Williams [1], Tracy Power [2], Jamie Stoneham [3], Nicole DeGreg [2] and Robert M. Siegel [1,4,*]** 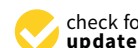

[1]   Heart Institute, Cincinnati Children's, Cincinnati, OH 45229, USA; jennawilliams422@gmail.com
[2]   Cincinnati Public Schools, Roberts Academy: A Paideia Learning Community, Cincinnati, OH 45215, USA; PowerTr@cpsboe.k12.oh.us (T.P.); Nicole.DeGreg@cincinnati-oh.gov (N.D.)
[3]   FarmChef, LLC, Cooking for the Family Program, Cincinnati, OH 45215, USA; jamie@thefarmchef.com
[4]   College of Medicine, University of Cincinnati, Cincinnati, OH 45229, USA
*   Correspondence: bob.siegel@cchmc.org; Tel.: +1-513-636-9420

**Abstract:** Roberts Academy is an urban elementary school consisting of primarily Hispanic students from lower socioeconomic homes. We were unable to provide weight management and healthy lifestyle counseling for many of the families that were referred to our obesity program, and more broadly were missing the at-risk families. The purpose of the Fit Kit intervention was to improve the health behaviors of the entire family at home and to reduce barriers through this comprehensive treatment and prevention approach. A Fit Kit included a shelf-stable, low-cost, healthy meal for a family of four, a portioned plate, and a soccer beach ball. Families also received monthly educational materials, healthy meal recipes, information about community food and exercise resources, and were invited to attend a culturally adapted cooking experience. No significant differences were seen between pre- and post-intervention survey questions. Post-Fit-Kit surveys showed that the majority of families prepared and liked the meal, found the recipes helpful, and used the portioned plate supplied with the kit. Overall, we conclude this is a feasible method for distributing tangible educational tools to families, but need more information about how it impacts food selection and purchasing behaviors of those families. While the Fit Kit proved to be feasible and acceptable in this test of concept, future studies are needed to further evaluate its impact in a more rigorous scientific manner.

**Keywords:** obesity; elementary school; healthy eating

## 1. Introduction

Childhood obesity is a major public health issue both at the national level and locally in Cincinnati, Ohio. It is estimated that approximately one third of American children are overweight or obese [1] and the prevalence of overweight and obesity for children in the greater Cincinnati area is higher than national childhood obesity prevalence estimates [2]. The obesity epidemic also disproportionately affects Hispanic and non-Hispanic black children, yielding increased comorbidities in this population [1,3].

Families who are knowledgeable and have confidence in their kitchen skills have greater intake of fruits and vegetables and overall improved diet quality [4–6]. Inadequate consumption of fruit and vegetables, excessive intake of sugary beverages, and low fiber intake are primary dietary factors that, when combined with inadequate physical activity, have been shown to lead to higher body mass index and weight in children and adolescents [3]. It is well documented that when families immigrate to America, fruit and vegetable consumption and physical activity levels often decrease [7–10]. Multiple explanations have been given for changes in food provided in the home, such as lack of money to

buy healthy/fresh food, lack of fresh/healthy food available during the year, lack of a variety of fruits and vegetables, and changing family preferences for unhealthy food [7]. Many immigrant families also view all non-perishable foods (including fruits and vegetables), which are more economical purchases during non-growing seasons, as unhealthy, since many are accustomed to fresh produce year-round [8,10].

In addition, food insecurity adds another barrier to healthy eating and ultimately can lead to obesity. Children and families experiencing food insecurity are more likely to consume nutrient-poor foods which have been correlated with chronic diseases such as obesity, diabetes, heart disease, and hypertension [11]. As with obesity, food insecurity disproportionately affects Hispanic and non-Hispanic blacks, and those who are unable to access safety-net programs [12,13]. These immigrant families have also been shown to have limited knowledge of other food resources, either private or community [13], and could benefit from information about what is accessible in their area for which documentation is not required. Schools across the United States have initiated programs to help reduce hunger at home for food-insecure families, with one of the most popular and widespread being backpack programs. Backpack programs send calorie-dense, easy to prepare foods home each weekend with potentially food-insecure students whose family income is below the poverty level. Although these programs have resulted in a decrease in food insecurity, the backpacks often contain food items that have high amounts of added salt and saturated fat, and are typically lacking in whole grains and other basic nutrients [14]. Passive education provided with the backpacks has been shown to decrease food insecurity [15], but testing whether education elicits behavior change around food selection had not been tested at the time of this study.

The Cincinnati Health Department has a free school-based clinic at Roberts Academy that treats students of the school, their families, and community members in the Price Hill neighborhood, regardless of their ability to pay or their immigration status. The school (preschool–8th grade) has approximately 809 students and the demographic breakdown is primarily Hispanic and African American, 61% and 31% respectively. Annually, the clinic treats about 483 children, and 44% are overweight or obese [16]. Unfortunately, due to the large volume of patients being seen at the clinic, the amount of time that the nurse practitioner can spend with each patient and family to discuss healthy eating, active living, and resources available in the community is limited. The need for additional time dedicated to specialized counseling around overweight/obesity for the majority of the families has been noted, and the Center for Better Health and Nutrition (CBHN)/HealthWorks! team at Cincinnati Children's—comprised of a pediatric obesity medicine specialist, registered dietitian, registered nurse, and exercise physiologist—was contacted about providing a school clinic service each month at Roberts Academy.

The CBHN/HealthWorks! team provides intensive group nutrition classes and exercise classes to teach both children and families about healthy eating and physical activity. Although the addition of the CBHN/HealthWorks! clinic means that more patients/families can be counseled than before, only 16% (32/200) of the students referred were able to participate in the CBHN/HealthWorks! program annually due to limited staffing and time for school clinics each month. Therefore, there was an immediate need for a large-scale intervention to reach the many families with children who are overweight and obese, as well as those families who have children who are at risk for overweight and obesity (not included in the referral system).

The Fit Kit program outlined here is a broad-stroke approach to reach all students and families with education and tangible teaching tools around healthy eating and active living. This program goes beyond the clinical approach by also providing culturally appropriate cooking classes that teach cost-effective meals that could be prepared at the homes of our target population (primarily Hispanic and non-Hispanic black families). The purpose of this study was to describe the implementation and test of concept of this intervention that could address food insecurity and poor lifestyle behavior.

## 2. Methods

### 2.1. Participants

Roberts Academy includes grades pre-Kindergarten through eighth grade, and all students enrolled during the 2017–2018 school year, regardless of weight status, and their families were targeted with the Fit Kit Intervention (*n* = 809).

### 2.2. Instruments

This pilot program was designed as a feasibility study, but program evaluation using surveys and informal information gathering at the cooking classes were also done to better understand satisfaction with the program components. Surveys were collected at three points during the program: a pre-intervention survey (September 2017), a post-Fit-Kit survey (November 2017), and a post-intervention survey (April 2018).

(1) 5-2-1-0 [17] Pre/Post Intervention Surveys (see questions presented in Table 1 and Table 2): Pre-intervention surveys sent home with students were the method used to obtain baseline data from families about their eating and physical activity behaviors, as well as their knowledge of community resources. Students returned the completed surveys to their teachers and a CCHMC staff member collected them from Roberts Academy for analysis. The same survey was sent home with students after the intervention at the end of the 2017–2018 school year. Students who returned completed surveys were entered into a draw for two bicycles at the end of the year. The survey used was adapted from 5-2-1-0 Let's Go! Maine Resources [17]. While not subject to scientific validation, the survey has been in use for over a decade.

(2) Satisfaction Survey about Fit Kit (see questions presented in Table 3): The month following the Fit Kit distribution, satisfaction surveys were sent home with students and collected as outlined above with the pre-/post-intervention surveys. Students who returned completed surveys were entered into a draw for two bicycles at the end of the year. This survey was developed specifically for this intervention and was not scientifically validated.

(3) Information Gathering: Before and after the cooking class, responses to questions were compiled from families around their current diet beliefs and behaviors at home.

**Table 1.** Results from Mann–Whitney Test of Pre-Intervention and Post-Intervention Survey Responses about Healthy Eating and Activity.

|  | Pre-Intervention Average (*n* = 186) | Post-Intervention Average (*n* = 64) | *P*-Value |
|---|---|---|---|
| How many fruits and vegetables does your child eat each day? | 3.44 | 3.40 | 0.884 |
| How many times does your child eat dinner at the table with the family each week? | 6.24 | 6.05 | 0.604 |
| How many times a week does your child eat fast food or takeout? | 1.34 | 1.23 | 0.555 |
| How many hours of screen time (television, computer, phone, tablet) does your child get each day at home? | 1.74 | 1.56 | 0.491 |
| How many minutes of physical activity does your child get at home | 64.23 | 69.30 | 0.862 |

**Table 2.** Results from Chi-Square Test of Pre-Intervention and Post-Intervention Survey Responses about Healthy Eating and Activity.

| | Pre- Intervention | | Post- Intervention | | Chi-Square Analysis |
|---|---|---|---|---|---|
| | Yes *n* (%) | No *n* (%) | Yes *n* (%) | No *n* (%) | *P*-Value |
| My child has a TV or Internet-connected device in his/her bedroom. | 65 (35) | 121 (65) | 15 (23) | 49 (77) | 0.12 |
| My child attends an after-school program at school or Cincinnati Recreation Center. | 34 (19) | 149 (81) | 11 (17) | 53 (83) | 1.00 |
| My child drinks juice, soda, Kool-Aid, or punch | 114 (63) | 66 (37) | 35 (55) | 29 (45) | 0.29 |
| Do you ever worry about running out of food before the end of the month? | 45 (24) | 141 (76) | 11 (17) | 53 (83) | 0.16 |
| Are you aware of the food banks and food pantries in your area? | 78 (43) | 105 (57) | 25 (40) | 37 (60) | 0.86 |

**Table 3.** Results from Post-Fit-Kit Survey Responses about Usage and Satisfaction.

| Question | Yes (%) | No (%) |
|---|---|---|
| My family ate the food from the Fit Kit provided. | 73 (91) | 7 (9) |
| My family liked the food provided in the Fit Kit. | 71 (89) | 9 (11) |
| The recipes provided were helpful. | 68 (85) | 12 (15) |
| My child uses the portioned plate. | 66 (83) | 14 (17) |
| My child played with the soccer beach ball. | 65 (81) | 15 (19) |

*2.3. Procedure*

The Fit Kit intervention had three components, as outlined below.

| Intervention Component | Description |
|---|---|
| Monthly educational materials | Monthly educational flyers (in English and Spanish) were sent home with all students around 5-2-1-0 messaging [17] during the 2017–2018 school year. Topics covered in handouts included tips for smart snacking, limiting sugary drinks, the importance of whole grains and fiber, samples of healthy breakfasts, tips to sleep better, limiting screen time, and exercise guidelines. |
| A one-time Fit Kit distribution to all students | The Fit Kit is designed to provide tangible tools for families to use at home to jumpstart healthy eating and activity behaviors. A Fit Kit costs about $12.00 total and includes a shelf-stable, low-cost, healthy meal for a family of four, a portioned plate to help with appropriate serving and portion sizes, recipes, and a soccer beach ball with game ideas. The goal of providing a Fit Kit instead of educational handouts alone is to reduce barriers (food insecurity, illiteracy, and lack of exercise equipment) for families as they learn about healthy eating and active living. Fit Kits were distributed to all students in October 2017. |
| Two family-focused cooking classes at the school | Partnered with the "Cooking for the Family," a 5 week culinary program where participants learn how to cook healthy and affordable meals for the family using fresh meat, grains and produce. One series was held in the fall of 2017 and one series was held in the spring of 2018. |

A "Healthy Meal" and "Healthy Eating" were defined as a meal and eating pattern that conformed to US Department of Agriculture standards as determined by the staff registered dietitian. As this was an observational study of a school-based intervention serving as a test of concept of the Fit Kit intervention, no power calculation was done prior to the implementation of the program.

*2.4. Data Analysis*

Number of Fit Kits distributed was recorded in October 2017. Data from the 5-2-1-0 pre- and post-intervention and Fit Kit satisfaction surveys were entered into Excel spreadsheets on a secure hospital computer. Descriptive aggregate statistics were determined for 5-2-1-0 pre- and post-intervention surveys and the satisfaction survey. Since the 5-2-1-0 survey was collected before and after the intervention, a Mann–Whitney U test was performed on the response to the first five survey questions (integer/non-parametric values) and chi-square testing was done on the last five questions (yes/no response) to determine whether any statistically significant changes occurred (Medcalc v 18.5; MedCalc Software, Ostend, Belgium).

## 3. Results

Eight educational handouts were distributed monthly to all students at Roberts Academy (*n* = 809) during the course of the intervention. Fit Kits were assembled and distributed to all 809 students in October 2017 as well. Eighteen families participated in the October cooking class series and 10 participated in the April cooking class series. Data were collected via feedback questionnaires and anecdotally from informal conversations between cooking class instructors and the families. Parents reported learning things in the "Cooking for the Family" program like how to hold a knife properly, new cooking techniques, and how to mix flavors together that had not been tried before. Even those participants who cooked frequently built on their current skills in the classes and said that it fostered a great community for learning.

Comparison of pre-intervention survey responses (*n* = 186, 23% response rate) to post-intervention responses (*n* = 64, 8% response rate) showed no significant differences using cross-sectional data from this cohort around their healthy eating and activity behaviors (Tables 1 and 2). Our sample size appeared to be too small to show any significant differences in behavior. Specifically, analyzing the change in a question like, "My child drinks juice, soda, Kool-Aid, or punch," would require a sample size of 793 in both pre- and post-survey response, which would have been close to a 100% response rate in this cohort and is unlikely with community survey data collection.

Results, however, from the post-Fit-Kit surveys (*n* = 80, 10% response rate) showed that 91% of families prepared the meal in the Fit Kit, 89% liked the meal, 85% found the recipes helpful, 83% used the portioned plate supplied with the kit, and 81% used the soccer ball (Table 3).

## 4. Discussion

With our feasibility study, we demonstrated that this is an acceptable and innovative model for distributing tangible healthy eating and physical activity materials to families. It was effective at reaching students and families, regardless of their weight status, and the school-based program was free and open to all, eliminating transportation and access barriers as previously documented. While our intervention was only a pilot and did not demonstrate any significant changes in behavior, the majority of families used the Fit Kit components and liked the pre-packaged meals. In 2018, an evaluation of a Power Pack backpack program in Indiana schools showed via survey response that the food sent home impacted the purchasing behaviors of the family [18]. This further lends support to the goal of making the food sent home with children healthier, not just calorie-dense.

*Limitations*

There were several limitations to this study, including a small sample size and that this intervention was evaluated primarily using self-reported, cross-sectional, unidentifiable survey data. Areas for

future research include conducting this program in a larger, more diverse population for greater generalizability and strength of pre- and post-survey differences. Matching responses from pre- and post-survey data would also provide more information about how programs like this impact the behavior changes of individual families. Unfortunately, since the questionnaires were anonymous in our study, this was not possible. Additionally, the surveys in this study were not scientifically validated. Although the 5-2-1-0 survey has been in use for over decade, the validity and reliability of both surveys has not been established. Finally, we did not assess the amount of physical activity of the children in minutes of play, but only whether they used the soccer ball supplied in the kit, nor did we assess whether there was any weight status change in the children after the intervention.

## 5. Conclusions and Future Directions

Overall, we conclude that this model is a feasible method for the distribution of educational materials focused on healthy eating and active living education. The school champion helped to organize the delivery of food items to the school and about 15 volunteers were able to pack 809 Fit Kits in about 90 min. Sending Fit Kits home with students at dismissal allowed for reduction of "lost" Fit Kits and increased the likelihood that they would make it home to the family. Additionally, the vast majority of families liked the program and receiving tangible educational tools as evidenced by the survey responses. Thus, our intervention served as a useful test of concept of an intervention that potentially addresses both a gap in healthy lifestyle knowledge and food insecurity. While the Fit Kit proved feasible and acceptable in this test of concept, future studies are needed to further evaluate its impact in a more rigorous scientific manner. This could include the validation of surveys and ideally a randomized controlled trial testing the Fit Kit program with outcomes such eating behavior, lifestyle knowledge, and hunger.

## 6. Human Subjects Approval Statement

This study was considered exempt by the Institutional Review Board at Cincinnati Children's Hospital prior to initiation.

**Author Contributions:** J.M.W. and R.M.S. were involved with the design and implementation of the project, the analysis, and the writing and editing of the manuscript. T.P., J.S. and N.D. involved with the design and implementation of the project and editing of the manuscript. All authors have read and agreed to the published version of the manuscript.

**Funding:** This project was funded by the National Center for Research Resources and the National Center for Advancing Translational Sciences, National Institutes of Health, through Grant 8 UL1 TR000077-05 and by a charitable contribution from Ethicon Corporation. The content is solely the responsibility of the authors and does not necessarily represent the official views of the NIH.

**Conflicts of Interest:** The authors have no conflict of interests.

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
