# Peer review of "A Feasibility Study of a Fit Kit School-Based Intervention to Improve the Health of Students and Their Families"

_reports, doi:10.3390/reports3010004_

Round 1

Reviewer 1 Report

Please see attached Word Document for my review. Very interesting study with great potential benefit for the children and families in our communities!

Author Response

Reviewer 1. Responses

Dear Reviewer:

Thank for your kind words and all your excellent suggestions.   They will be addressed point by point.

Abstract

I recommend adding an “Implications and Future Directions” section to the manuscript. If this is added, add this to abstract.

Response: Future Directions has now been added to both the Abstract and Manuscript as a co-heading to what was previously labelled “Conclusions” and is now labelled “Conclusions/Future Directions.  This is in the abstract lines 27-29 and in the manuscript lines 198-200 with:

“While the Fit Kit proved feasible and acceptable in this test of concept, future studies are needed to further evaluate its impact in a more rigorous scientific manner.”

Introduction

Overall, the language used throughout the introduction (and throughout the

manuscript) is clear and easy to follow.

I appreciate the writers’ continual reiterating of the importance of improving education

and accessibility to nutritional meals and health-related information.

Response: Thank you for your kind words.

Spelling error is noted in line 51. Change “immigrant” to “immigrants” or state

“immigrant families.”

Response: This error has been corrected to “immigrant families.”

Consider providing information about common qualifications for backpack programs

when discussing these in lines 56-57.

Response: This has now been clarified with lines 58-60:

Backpack programs send calorie dense, easy to prepare foods home each weekend with potentially food insecure students whose family income is below the poverty level. 

Methods

Please include the reliability and validity of each of the measures used in this study, including the “5-2-1-0 Let’s Go! Main Resources” mentioned in line 106.

Response: Thank you for bringing this to our attention.  Neither survey used in this study has been scientifically validated, although the 5-2-1-0 survey has been in use for over a decade.  This is now clarified in the Methods lines 111-112 and 116-117 with:

“While not subject to scientific validation, the survey has been in use since for over a decade.” and

“This survey was developed specifically for this intervention and was not scientifically validated.”

This is also mentioned in the limitation section with lines 185-187:

“Additionally, the surveys in this study were not scientifically validated.  Although the 5-2-1-0 survey has been in use for over decade, the validity and reliability of both surveys have not been established.”

It appears that some of the results information is included in the Methods section.

consider moving Tables 1, 2, and 3 to the Results section.

Consider re-formatting the heading in Table 2. Consider re-aligning the “Yes n (%)” and

“No n (%)” headings to be on one line.

Response: Thank you for these suggestions.  The tables have been moved to the results section and headings of Table 2 have been reformatted.

Spelling error is noted in the Procedure section. “Cost” should be “costs.”

Response: Thank you for pointing out this error.  It has been corrected.

Consider explaining how the participants who “cook frequently” was determined after

mentioning this in line 137. How were the participants who cook frequently

differentiated from those who do not?

Response:  This observation was anecdotal and has been dropped from the manuscript.

In line 138, consider explaining how data were collected (e.g., survey response,

qualitative interview, etc.) when mentioning that participants said the classes fostered a

great community for learning. Was this included as an item on one of the surveys?

Response:  These observations were not through surveys or focus groups but anecdotally described by the cooking class instructors.  This is now clarified in lines 140-145

“Data was collected via feedback questionnaires and anecdotally from informal conversations between cooking class instructors and the families.   Parents reported learning things in the “Cooking for the Family” program like how to hold a knife properly, new cooking techniques, and how to mix flavors together that had not been tried before.  Even those participants who cook frequently built on their current skills in the classes and said it fostered a great community for learning.”

Spelling error noted in line 142. Change “and” to “any” if this was the intended word

choice.

Response:  Thank you for finding this error.  It has been corrected.

Discussion

In line 166, add a period to the end of the sentence.

Response:  This has been corrected.

Consider re-wording the first sentence under “Conclusions.” Perhaps consider changing it to “Overall, we conclude that this model is a feasible method for distribution of

educational materials focused on healthy eating and active living education.”

Response:  Thank you for the suggested wording.  This has been incorporated into the manuscript.

Consider operationally defining “healthy” as it is used throughout the manuscript. How

was “healthy” determined? Were nutritionists or dieticians consulted? This word can

mean different things to different people.

Response:  Very good point.  This is now defined on lines 123-124:

“A “Healthy Meal” and “Healthy Eating” were defined as a meal and eating pattern that conformed to US Department of Agriculture standards as determined by the staff registered dietitian.”

Consider explaining in lines 172-173 how the information was collected that families

liked the program and receiving tangible educational tools. Was this information

obtained via survey, interview, etc?

Response this is now clarified in lines 195-197 with:

“Additionally, the vast majority of families liked the program and receiving tangible educational tools as evidenced by the survey responses.”

Recommend adding a paragraph dedicated to discussion of recommended future

research based on your findings.

Recommend including a paragraph dedicated to reiterating implications of the study.

Response:  These are excellent recommendations and both have been added to the final section “Conclusions and Future Directions” with lines 190-202:

“Overall, we conclude that this model is a feasible method for distribution of educational materials focused on healthy eating and active living education.  The school champion helped organize the delivery of food items to the school and about 15 volunteers were able to pack 809 Fit Kits in about 90 minutes.  Sending Fit Kits home with students at dismissal allowed for reduction of “lost” Fit Kits and increased likelihood they would make it home to the family.  Additionally, the vast majority of families liked the program and receiving tangible educational tools as evidenced by the survey responses.   Thus, our intervention serves as a useful test of concept of an intervention that potentially addresses both a gap in healthy lifestyle knowledge and food insecurity.  While the Fit Kit proved feasible and acceptable in this test of concept, future studies are needed to further evaluate its impact in a more rigorous scientific manner.  This could include the validation of surveys and a ideally a randomized controlled trial testing Fit Kit program with outcomes such eating behavior, lifestyle knowledge and hunger.” 

References

The references should be listed in alphabetical order.

While we agree, there are some distinctive benefits to having the references listed in alphabetical order, the preferred format as describe by instructions to authors for Reports is numbered in order of appearance. 

Again, thank you for the careful review and the wonderful suggestions.

Reviewer 2 Report

The manuscript describes a fit kit intervention that was meant to improve the health behaviors of the family and address the issue of obesity/overweight among children. The study, however, showed no significant change post-intervention and thus the authors concluded that fit kit is a feasible intervention.

Did the authors do a sample size calculation? It is clear that the study was underpowered to detect any significant changes.  What was the study design? the results section stated that there was a qualitative component..how was it analysed and why the results are not presented? 81% of the children played with the soccer ball. The data would have been interesting if it showed a considerable increase in the hr of physical activity (eg. number of hrs the child played with the ball), any change in the weight pre-post intervention.  The study could have been interesting with a better planning and design. The study didn't achieve the aim it was designed for.

Author Response

Reviewer 2. Responses

Dear Reviewer:

Thank you so much for the careful and suggestions.  We address them point by point as follows:

·         The manuscript describes a fit kit intervention that was meant to improve the health behaviors of the family and address the issue of obesity/overweight among children. The study, however, showed no significant change post-intervention and thus the authors concluded that fit kit is a feasible intervention.

Response:  Thank for your comments and helpful suggests to improve our manuscript. 

·         Did the authors do a sample size calculation?

·          It is clear that the study was underpowered to detect any significant changes.  What was the study design?

Response: Thank you for bringing up these two points.  The study was an observational study of a test of concept.  Thus, no power calculation was done prior to implementation.  This is now explained more clearly in the Introduction and Methods sections in lines 90 to 92 and 125-126:

“The purpose of this study was to describe the implementation and test of concept of intervention that could address food insecurity and poor lifestyle behavior.”  and

“As this was an obeservational study of a school-based intervention serving as a test of concept of the Fit Kit intervention, no power calculation was done prior to the implementation of the program.”

·         the results section stated that there was a qualitative component..how was it analysed and why the results are not presented?

Response:  Thank you for asking for clarification.  The data was obtained by surveys and anecdotally from informal discussion with the class instructors.  This is now clarified in lines 140-141 with:

“Data was collected via feedback questionnaires and anecdotally from informal conversations between cooking class instructors and the  families.”  

·          81% of the children played with the soccer ball. The data would have been interesting if it showed a considerable increase in the hr of physical activity (eg. number of hrs the child played with the ball), any change in the weight pre-post intervention.  

Response: We agree these are outcomes that should be addressed in a more definitive study of the Fit Kit.  This is now described in the limitations section, lines 187-189 with:

“Finally, we did not access the amount of physical activity of the children in minutes of play, but only if they used the soccer ball supplied in the kit or access if there was any weight status change in the children before or after the intervention.”

·         The study could have been interesting with a better planning and design. The study didn't achieve the aim it was designed for. 

Response: We agree that our study has many limitations which are now more clearly defined in the limitations section.  Additionally, we should have been more clear that this was an observational study of a test of concept.  We believe that we have more clearly described this and taken it that context, we have met the aim of the study.  Thank you so much for the helpful critique and all the suggestions.

Round 2

Reviewer 2 Report

No further comments